# Hydroxychloroquine-Loaded Chitosan Nanoparticles Induce Anticancer Activity in A549 Lung Cancer Cells: Design, BSA Binding, Molecular Docking, Mechanistic, and Biological Evaluation

**DOI:** 10.3390/ijms241814103

**Published:** 2023-09-14

**Authors:** Fawzia I. Elshami, Hadeer A. Shereef, Ibrahim M. El-Mehasseb, Shaban Y. Shaban, Rudi van Eldik

**Affiliations:** 1Chemistry Department, Faculty of Science, Kafrelsheikh University, Kafrelsheikh 33516, Egypt; fawzyaalshami@yahoo.com (F.I.E.); elmehasseb2@gmail.com (I.M.E.-M.); 2Clinical Pathology Department, University Hospital, Menoufia University, Shebin El-Kom 32512, Egypt; hadeersherif35@gmail.com; 3Department of Chemistry and Pharmacy, University of Erlangen-Nuremberg, 91058 Erlangen, Germany; 4Faculty of Chemistry, Nicolaus Copernicus University in Torun, 87-100 Torun, Poland

**Keywords:** A549 lung cancer, nanoparticles, cell penetration, drug affinity, dissociation constants, molecular docking, antibacterial, hydroxychloroquine

## Abstract

The current study describes the encapsulation of hydroxychloroquine, widely used in traditional medicine due to its diverse pharmacological and medicinal uses, in chitosan nanoparticles (CNPs). This work aims to combine the HCQ drug with CS NPs to generate a novel nanocomposite with improved characteristics and bioavailability. HCQ@CS NPs are roughly shaped like roadways and have a smooth surface with an average size of 159.3 ± 7.1 nm, a PDI of 0.224 ± 0.101, and a zeta potential of +46.6 ± 0.8 mV. To aid in the development of pharmaceutical systems for use in cancer therapy, the binding mechanism and affinity of the interaction between HCQ and HCQ@CS NPs and BSA were examined using stopped-flow and other spectroscopic approaches, supplemented by molecular docking analysis. HCQ and HCQ@CS NPs binding with BSA is driven by a ground-state complex formation that may be accompanied by a non-radiative energy transfer process, and binding constants indicate that HCQ@CS NPs–BSA was more stable than HCQ–BSA. The stopped-flow analysis demonstrated that, in addition to increasing BSA affinity, the nanoformulation HCQ@CS NPS changes the binding process and may open new routes for interaction. Docking experiments verified the development of the HCQ–BSA complex, with HCQ binding to site I on the BSA structure, primarily with the amino acids, Thr 578, Gln 579, Gln 525, Tyr 400, and Asn 404. Furthermore, the nanoformulation HCQ@CS NPS not only increased cytotoxicity against the A549 lung cancer cell line (IC_50_ = 28.57 ± 1.72 μg/mL) compared to HCQ (102.21 ± 0.67 μg/mL), but also exhibited higher antibacterial activity against both Gram-positive and Gram-negative bacteria when compared to HCQ and chloramphenicol, which is in agreement with the binding constants. The nanoformulation developed in this study may offer a viable therapy option for A549 lung cancer.

## 1. Introduction

Lung cancer was the most prevalent and lethal cancer in the world in 2020, with 2.2 million diagnoses and 1.8 million deaths [1]. Many lung cancer therapy regimens are built on the platinum-based DNA-damaging drugs cisplatin and carboplatin. These medicines are cytotoxic to rapidly dividing cells, causing an overwhelming quantity of irreversible DNA damage and guiding cells toward apoptotic cell death [2,3]. The combined activities of a diverse array of DNA repair proteins keep the genetic code stable in normal, noncancerous cells. These proteins detect code damage, begin cell cycle checkpoint signaling, and either repair the lesion or, if it is too severe, activate apoptotic pathways. This regulation is gradually lost in malignant cells, resulting in an accumulation of mutations [4,5,6].

Recently, research has concentrated on the development of nanoscale-based drug delivery systems that control the release of chemotherapeutic drugs directly into cancer cells and change the way cancer is diagnosed and treated [7,8]. Because of their small size and larger specific area, nanoparticles absorb more drugs than larger drug carriers [9]. Chitosan (CS) is one of the most prevalent polysaccharides and polymers that have a variety of biological functions, including bioadhesion, physicochemical properties, and permeability-enhancing capabilities, making it a unique material for the development of drug delivery systems. CS nanoparticles (CS NPs) are a top choice for a drug delivery vehicle because they have better intracellular accumulation, controlled release capabilities, improved drug delivery, therapeutic efficacy, and a protective barrier after encapsulating medicines [10,11,12,13,14]. A number of publications have already proven the anticancer activity of CS NPs loaded with drugs against lung cancer A549 cells [15,16,17,18]. Unfortunately, many tumors respond to chemotherapy but eventually acquire resistance [19], in addition to the negative side effects [20]. Finding novel, efficient, and non-toxic molecules from natural resources is more crucial than ever as alternative cancer treatments [21,22,23].

Hydroxychloroquine (HCQ) (Figure 1) has been an antimalarial drug used to treat disorders other than malaria since 1894, when Payne reported that quinine was effective in cutaneous lupus [24]. HCQ has therapeutic effects in a variety of different disorders [25], and recently, it has been used off-label as a potential therapy for COVID-19, a new coronavirus [26]. HCQ has been used in anticancer therapy, particularly in combination with traditional anticancer treatments, since it can sensitize tumor cells to a variety of drugs, increasing the therapeutic action. Surprisingly, HCQ has impacts on both cancer cells and the tumor microenvironment. HCQ inhibits the autophagy flux, which is the best researched anticancer action, as well as the toll-like receptor 9, p53, and CXCR4-CXCL12 pathways in cancer cells. Additional clinical research suggested that HCQ merits additional clinical research in a variety of cancer types [27,28,29,30,31,32,33,34,35]. Recently, HCQ was found to have an inhibitory effect on cellular autophagy, is a lysosomal inhibitor, and therefore has potential application in synergy with chemotherapeutic drugs [36,37]. For HCQ, the choice of biomaterial affects the delivery system’s biocompatibility, different physicochemical characteristics, and capacity to release precisely within the cell. As a result, we synthesized HCQ@CS NPs for this investigation, where the chemotherapeutic drug HCQ was encapsulated in the smaller nanoparticle CS NPs. The encapsulation of HCQ in CS NPs is expected to result in a new nanocomposite with improved properties and biological results. We described their chemical and physical characteristics, such as particle size, zeta potential, scanning electron microscopy (SEM), and Fourier transform infrared spectroscopy (FT-IR), and their anticancer efficacy against the A549 lung cancer cell line was evaluated.

Cancer patients are frequently given a multi-drug cocktail of antibacterial agents in addition to chemotherapeutic agents to prevent various infections because chemotherapy actually makes cancer patients immunocompromised and susceptible to infections [38]. For this reason, the antibacterial properties of HCQ@CS NPs and HCQ were also tested in vitro against four bacteria strains (*E. coli*, *P. aeruginosa*, *S. aureus*, and *E. faecalis*) using the dilution tube method (DTM) technique.

It is crucial to determine whether or not a novel anticancer metalotherapeutic binds to serum transport proteins because intravenous administration is the preferred method for administering such drugs. In fact, examining a novel candidate drug’s interactions with significant (the most prevalent) serum proteins is necessary for its pharmacological characterization. As a result, it is necessary to calculate the binding stoichiometry, affinity constant, and number of (particular) binding sites. Human serum albumin (HSA), immunoglobulin (G), and serum transferrin (Tf) are the three main serum proteins. The medicinal chemistry literature frequently describes interactions between BSA and prospective metallodrugs. This is due to BSA’s stability, neutrality in a wide range of metabolic processes, low cost, and structural closeness to its human analog, HSA [39,40] and references therein]. Using steady-state fluorescence and UV–vis spectroscopy, the binding of BSA to HCQ and HCQ@CS NPs was explored. We used a stopped-flow technique for carrying out a kinetic investigation of the binding process because it is thought that DNA/protein–drug interaction is a fast step. This method aids in identifying the kinetic characteristics of the phases leading to intermediate species and ultimately obtaining an interaction mechanism [41,42,43,44,45,46], The geometrically optimized parameters (bond lengths, bond angles, and dihedral angles) were computed using DFT/B3LYP theory on the chitosan moiety and HCQ. Docking simulation will be used in order to confirm the spectroscopic results of the binding of HCQ to the BSA structure and the formation of the HCQ–BSA complex.

## 2. Results

### 2.1. Synthesis and Characterization

By deacetylating chitin, chitosan is produced (Figure 1); during this process, certain N-acetylglucosamine moieties are changed into glucosamine units. Since chitosan’s pKa value is close to 6.5, its solubility in acidic aqueous conditions can be attributed to the structure’s presence of protonated -NH_2_ groups [47]. Chitosan dissolves when approximately 50% of all amino groups are protonated [48]. Free amine groups can be protonated in an acidic medium and ionically bond with the reactive groups of the drug molecules via an encapsulation strategy to form new compounds, while free hydroxyl groups make up the skeleton of chitosan and are easily capable of hydrogen bonding with other molecules [49]. HCQ contains a hydroxyl group at the end of the side chain; the N-ethyl substituent is β-hydroxylated. By taking advantage of the noncovalent interactions such as the hydrogen bond between the positively charged amino groups in CS and the hydroxyl group of HCQ, smaller NPs loaded with HCQ were first made using the ionic gelation method. Then, using sodium tripolyphosphate (TPP) as the emulsifier, the O/W emulsion method was used to incorporate HCQ into CS. This approach allows for the formation of larger NPs that are loaded with HCQ. SEM observation of HCQ@CS NPs reveals that the HCQ-loaded CS nanoparticles have a smooth surface with a diameter of about 500 nm. The field of view of HCQ@CS NPs at 500 nm is depicted in Figure 2a.

Dynamic light scattering (DLS) was used to determine the specific size distribution and potential of HCQ and HCQ@CS NPs. The mean particle sizes of HCQ@CS NPs (159.3 ± 7.1 nm) and HCQ (255.7 ± 18.45 nm) with polydispersity index (PDI) values of 0.224 ± 0.101 and 0.337 ± 0.083, respectively, are shown in Table 1 and Figure 2b and are like those found under SEM, with appropriate dimensions and good dispersion. The size of the NPs was one of many factors that affected the EPR effect, with particles having a diameter of less than 200 nm being the most effective and a cut-off size of about 400 nm for tumor penetration [50]. The zeta potential (ζ) of HCQ@CS NPs and HCQ is also characterized. The zeta potential of HCQ@CS NPs was +46.6 ± 0.8 mV, and that of HCQ was −16.4 mV (Table 1 and Figure 2c). Higher anticancer activity resulted from better tumor cell absorption of the positively charged nanodrugs and their potential to escape from lysosomes after cell entry [51].

The current study used FT-IR spectroscopy to chemically characterize HCQ@CS NPs by comparing the spectra of HCQ@CS NPs with those of HCQ, and Figure 2d depicts possible interactions. The characteristic peaks in the IR spectra of CS can be seen at wavelengths of 1150 cm^−1^ for C–O–C stretching, 1465 cm^−1^ for C–H stretching, 3514 cm^−1^ for O–H stretching, and 1596 cm^−1^ for N-H stretching. The HCQ data also demonstrate distinctive N-H peaks at 3381 cm^−1^ and 3100 cm^−1^, as well as three additional peaks at 1615 cm^−1^, 1551 cm^−1^, and 1470 cm^−1^ that point to the presence of an aromatic ring structure. Some of the absorption peaks lose strength or vanish entirely after the fabrication of HCQ@CS NPs, and the wave number changes. For instance, the stretching vibration shift of -OH and -NH_2_ at 3434 cm^−1^ to 3381 cm^−1^ and the stretching shift of the amide group at 1657 cm^−1^ to 1615 cm^−1^ are seen in the FT-IR spectra of the final NPs. These alterations imply that the synthesis of HCQ@CS NPs is a chemical reaction between HCQ and CS NPs rather than a physical combination, indicating that the HCQ pharmaceuticals are successfully contained in the finished NPs [52].

### 2.2. Computational Finding

All theoretical calculations were performed with the Gaussian 09 W package [53] using density function theory (DFT) [54,55]. The optimized structures of HCQ, monomer of chitosan (M), and dimer of chitosan (MM) were employed using Becke 3-Parameter, Lee, Yang, and Parr (B3LYP) [56,57] with a 6–31G(d) basis set. All calculations were perfectly optimized in the gas phase. Frequency calculations for each optimized geometry were done at the same theoretical level to verify that they are all minima or transition states on the system’s potential energy surface. Figure 3 depicts the optimized geometries of the investigated HCQ, M, and MM. The bond lengths on HCQ; C_1_–N_1_, C_14_–N_2_, C_3_–N_3_, H_26_–N_3_, C_4_–C_5_, C_6_–Cl_1_, C_17_–O_1_ and O_1_-H_22_ are 1.354, 1.474, 1.442, 1.022, 1.458, 1.781, 1.427 and 0.97 Å, respectively, when the bond angles on HCQ; C_2_–C_3_–C_4_, C_4_–C_5_–C_8_, C_5_–C_6_–C_7_, C_17_–O_1_–H_22_, C_1_–N_1_–C_5_ and C_11_–N_2_–C_16_ are 118°, 121°, 123°, 107°, 117° and 113°, respectively. The bond lengths on M; C_4_–C_3_, C_5_–O_1_, C_4_–O_1_, C_2_–O_4_, O_4_–H_8_, C_3_–N_1_, N_1_–H_12_ and C_4_–H_4_ are 1.58, 1.44, 1.37, 1.38, 0.97, 1.38, and 1.01 Å, respectively, when the bond angles on M; C_2_–C_1_–C_5_, C_1_–C_2_–C_3_, C_5_–O_1_–C_4_, C_2_–O_4_–H_8_, C_3_–N_1_–H_13_ and H_12_–N_1_–H_13_ are 113°, 107°, 116.5°, 111.35°,119° and 118°, respectively. The bond lengths on MM; C_4_–O_1_, C_5_–O_1_, C_7_–O_4_, C_11_–O_4_, C_1_–O_8_, C_11_–O_8_, C_5_–C_6_, C_10_–C_11_, C_4_–H_2_, and C_10_-N_2_ are 1.4, 1.44, 1.44, 1.42, 1.44, 1.37, 1.53, 1.57, 1.09, and 1.42 Å, respectively, when the bond angles on MM; C_2_–C_1_–C_5_, C_5_–O_1_–C_4_, C_1_–O_8_–C_11_, C_2_–O_9_–H_24_ and C_3_–N_1_–H_8_ are 112°, 116°, 121°, 106°, and 117°, respectively. These optimization structures are in agreement with the results reported by Onoka et al. [57].

#### 2.2.1. Quantum Chemical Parameters

The distribution of the highest occupied molecular orbital (HOMO), lowest unoccupied molecular orbital (LUMO), and frontier molecular orbitals (FMO) in HCQ, M, and MM was performed using the ChemCraft program [58]. The HOMO and LUMO explain the electronic transition of molecules and also indicate the electrophilic and nucleophilic attraction of compounds. Additionally, the chemical reactivity, stability, and hardness of the compounds are explained by the energy gap, defined as the difference in energy between HOMO and LUMO. A hard molecule is one with a large HOMO–LUMO gap, whereas a soft molecule with high reactivity is one with a shorter HOMO–LUMO gap. While softness gauges chemical reactivity, hardness gauges molecule stability. From the data of HOMO and LUMO energies, the ionization potential (IP), electron affinity (EA), the energy gap (ΔE) = (IP−EA), absolute electronegativity (χ), electronic chemical potentials (Pi), absolute hardness (η), absolute softness (σ), global softness (S), and global electrophilicity (ω) are presented in Table 2. The global reactivity can be defined using Koopman’s theorem (Appendix A).

The data, depicted in Table 2, show the following points of association between the compounds: (i) The energy gaps of M in comparison with MM reflect the high softness and biological activity; (ii) In M and MM, χ is positive (+6.67 and +5.97, respectively), whereas Pi has negative values (−6.67 and −5.97, respectively), indicating that the molecule is able to capture electrons from its environment and that its energy must decrease when accepting the electron charge. In HCQ, χ is negative (−2.44), whereas Pi is positive (+2.44), indicating that the molecule is able to donate electrons to its environment and that its energy must increase when donating the electron charge.

#### 2.2.2. Frontier Molecular Orbitals and Chemical Reactivity

Frontier molecular orbitals (FMO) are used to understand several types of reactions and to predict the most reactive position in a molecular orbital and its properties. The HOMO level is the highest energy orbital containing electrons, which acts as an electron donor, while the LUMO level is the lowest energy orbital, which acts as an electron acceptor. The calculated HOMO of HCQ is located on C_1_, C_2_, C_3_, C_6_, C_7_, C_8_, C_9_, C_12_, N_1_, N_3_, and Cl_1_, whereas the LUMO of HCQ is located on C_16_, C_17_, O_1_, and N_2_. The HOMO of M is located on C_5_, O_1_, O_5_, and H_9_, whereas the LUMO is located on C_2_, C_3_, O_2_, O_3_, O_4_, N_1_, and H_13_. The HOMO level of MM is located on C_7_, C_9_, O_3_, O_4_, O_5_, O_6_, O_7_, and O_8_, whereas LUMO is located around all nitrogen atoms, all oxygen atoms except O_3_, and all carbon atoms except C_6_ or the overall skeleton except all atoms of hydrogen, O_3_, and C_6_ (Figure 4).

Molecular electrostatic potential (MEP) is a very suitable tool for highlighting reactive sites against electrophilic and nucleophilic attack and determining the relative polarity of molecules. The MEPs of HCQ, M, and MM are presented in Figure 5. The red and yellow sites (negative regions) of MEP are related to electrophilic reactivity, while the blue and green sites (positive regions) are related to nucleophilic reactivity. Figure 5 reveals that the negative MEP regions of HCQ are concentrated on two rings (around the aromatic site), thus confirming their electron donor ability (electrophilic reactivity), but the M and MM appear as blue and green (positive regions), thus confirming their electron acceptor ability (nucleophilic reactivity).

### 2.3. BSA Binding Studies

#### 2.3.1. Fluorescence Studies

The most prevalent protein in mammal blood is BSA. Its main job is to deliver ions and medications to cells and tissues [59,60]. Due to its significant function, it is necessary to investigate the potential for biologically active compounds and BSA to interact. This could result in the biological properties of the biologically active compound being lost or enhanced, or it could reveal distinct routes for potential drug delivery [61]. As is well known, when excited at 295 nm, the solutions of free BSA exhibit a strong fluorescence emission band with λ_em,max_ = 343 nm (due to the presence of two tryptophans at positions 134 and 212). Fluorescence emission spectroscopy, more especially the quenching of the associated emission bands, can be used to observe any potential interactions between physiologically active substances and BSA [59,60]. In the presence of HCQ and HCQ@CSNPs, the fluorescence emission spectra of BSA showed strong quenching of the corresponding BSA emission band (Figure 6); the initial BSA fluorescence emission (*F*/*F*_0_) was quenched by up to 75% for HCQ@CSNPs and 50% for HCQ. The quenching in the fluorescence emission spectra of the BSA brought on by the presence of the complexes may be attributed to modifications in the tryptophan environment of the BSA, most likely brought on by modifications in the secondary structure of albumin, and may serve as an indirect sign of the complexes’ binding to the BSA [62].

Figure 6a shows the fluorescence spectra of BSA with increasing concentrations of HCQ@CSNPs, and the fluorescence intensity at 338 nm (tryptophan) is decreasing continuously along with a blue shift of about 30 nm when excited at 280 nm. This is an indication of the binding of HCQ@CSNPs with BSA, which causes a change in the microenvironment of tryptophan residue. Local polarity around the tryptophan residue must have increased, causing a change in the compact structure of the hydrophobic sub-domain where tryptophan was placed (Trp-212). This local perturbation at the IIA binding site in BSA results in conformational change whereby Trp 134 is more exposed to polar environment, which explains the red shift in BSA fluorescence. Thus, HCQ@CSNPs nanocomposite-induced conformational changes on BSA can lead to increased association among BSA moieties, and this self-quenching may also be a probable reason for the decreased fluorescence intensity and blue shift. On the other hand, the fluorescence spectra of BSA with increasing concentrations of HCQ showed a continuously decreasing intensity with no shift when excited at 280 nm. The determination of the BSA-quenching constants (*k*_q_) and the BSA-binding constants (*K*_SV_) by the Stern–Volmer and Scatchard equations and the corresponding plots were used to further study the interaction of HCQ@CSNPs and HCQ with BSA. More specifically, the Stern–Volmer equation (Equation (1) [44]) and Scatchard equation (Equation (2) [50]) were used to compute the values of the *k*_q_ and *K* constants of HCQ@CSNPs and HCQ (Table 3), where the value *τ*_o_ = 10^−8^ s was used as the fluorescence lifespan of tryptophan in the BSA [62].
(1)$$\frac{F_{0}}{F}=\left(1+K_{SV}\left[Q\right]\right)=1+k_{q}\tau_{0}$$
(2)$$\frac{\frac{∆F}{F_{0}}}{\left[Q\right]}=nK-K\frac{∆F}{F_{0}}$$
where *F*_0_ and *F* denote fluorescence intensities in the absence and presence of quenchers, respectively. *K*_SV_ is the Stern–Volmer quenching constant, and [*Q*] is the concentration of the quencher. *τ*_0_ is the average fluorescence lifetime of BSA, as given in Table 3. *k*_q_, which equals *K*_SV_/τ_o_, is the quenching rate constant of the biomolecule, and τ_o_ the average lifetime of the biomolecule in the absence of a quencher.

The plot of *F*_0_/*F* versus [Q], which gives the *K*_sv_ values as slope, and the values of *K*_q_ are calculated by using Equation (1), and these results are given in Figure 6 and summarized in Table 2. The plots exhibited a linear relationship, which indicated only one type of quenching (static or dynamic quenching). The plot of $\frac{∆F}{F_{o}[Q]}$ versus $\frac{∆F}{F_{0}}$ gives the *K* values, as slope as shown by Scatchard Equation (2) (Figure 7). The quenching constants are relatively high (∼10^4^ M^−1^) with HCQ-CSNPs exhibiting the highest K values. The bimolecular quenching constants, *k*_q_ value was of the order of 10^12^ M^−1^ s^−1^. The considerably high *k*_q_ values were probably an indication of a static quenching process via complex formation [59,63].

#### 2.3.2. Determination of Binding Constant (K_bin_), Number of Binding Sites (n) and ΔG

The double logarithmic Equation (3) was used to further analyze the fluorescence data, and the results allowed for the determination of the binding constant (*K*_a_) and binding stoichiometry (*n*) of the drug/BSA system. The linear dependence of log(*F*_0_/*F*)/*F* on log [Q] is seen in Figure 8a. The slope was used to calculate the number of binding sites (*n*) for the molecule, while its intercept (log*K*_a_) provided information on the binding constant [64,65,66,67,68,69].
(3)$$log\left(\frac{F_{0}-F}{F}\right)=logK_{bin}+nlog[Q]$$

The values of *n* for the BSA drug system were close to one, implying that the drug occupies only one binding site in BSA. The binding constant *K*_a_ of magnitude 10^−4^ M^−1^ implies that BSA is strongly bound. Equation (4) uses the value of the binding constant *K*_a_ to calculate the free energy change, ΔG, of drug binding to BSA [70]. The negative free energy change (Table 3) implies that the interaction with BSA is thermodynamically favorable.
Δ*G* = −*RT* ln *K*_a_(4)

#### 2.3.3. Determination of Accessible Fraction of the Fluorophore

When the quencher (HCQ@CSNPs and HCQ) is added, the BSA fluorescence intensity decreases (Figure 6), indicating that the quencher has bound to a location close to the BSA fluorophore tryptophan 213 (Trp 213). It is because the BSA was excited at 278 nm to prevent tyrosine residues from contributing. In accordance with the concept, a fluorophore can only be quenched by a quencher when both the fluorophore and the quencher are nearby. Therefore, the observed quenching by HCQ@CSNPs and HCQ shows that these molecules bind to BSA near Trp 213. The modified Stern–Volmer Equation (5) was used to calculate the extent of fluorophore accessibility by HCQ@CSNPs and HCQ [64,71].
(5)$$\frac{F_{0}}{F_{0}-F}=\frac{1}{f_{a}K_{SV}[Q]}+\frac{1}{f_{a}}$$
where *K*_sv_, the Stern–Volmer quenching constant, *f*_a_, the proportion of the fluorophore that is accessible to the quencher, and Δ*F*(*F*_0_ − *F*), the change in fluorescence intensity, are all included in Equation (5). To fit the quenching data to Equation (5), a graph between *F*_0_/Δ*F* and 1/[Q] was plotted (Figure 7b). The *K*_sv_ value estimated from the intercept to slope ratio is in the order of 10^4^, which is the same order as the *K*_sv_ values determined by the Stern–Volmer plot. The intercept’s measured fraction of fluorescence (*f*_a_) accessible to the quencher (Table 3) demonstrates the fluorophore’s complete accessibility to the quencher. As a result, the fluorophore’s complete accessibility provides additional evidence that HCQ@CSNPs and HCQ bind close to Trp 213.

#### 2.3.4. Conformation Investigation Using UV−Vis Spectroscopy

The two most prevalent methods for quenching fluorescence are static and dynamic quenching, which differ in their disruption effects on the fluorophore’s absorption spectrum. It has been found that dynamic quenching has no effect on the absorption spectrum of fluorophores; nevertheless, static quenching has an effect on the fluorophore spectrum [72,73,74,75]. The UV–vis spectra of BSA were recorded at RT (25 °C), and the effect of HCQ and HCQ@CSNPs on the absorption spectrum was analyzed to gain insight into the method involved in quenching the fluorescence of BSA by HCQ and HCQ@CSNPs. Figure 9a shows that when HCQ@CSNPs were added, the absorbance at 250 nm reduced considerably, accompanied by a blue shift of 5 nm, indicating static quenching and structural modification in the protein secondary structure. It was also found that the intensity of the 280 nm peak, which is related to the π–π* transition of Trp, Tyr, and Phe, is somewhat increased, implying that the microenvironment around these amino acids has changed [76]. The emergence of isosbestic points at 264 and 287 nm, in addition to the alterations at 250 and 280 nm, shows the development of an equilibrium binding system of HCQ@CSNPs and BSA. The absorbance at 250 nm was found to considerably increase with no discernible shift after the addition of HCQ, which suggests static quenching with only minor changes to the protein secondary structure. Additionally, a minor rise in the strength of the 280 nm peak, which is connected to the Trp, Tyr, and Phe π–π* transition, suggests that the microenvironment surrounding these amino acids has changed. To calculate the binding constant, the change in absorption data at 224 nm was fitted to the double reciprocal Equation (6) [76,77,78]. The binding constant (*K*_b_) was 0.4 × 10^4^ M^−1^ and 0.82 × 10^4^ M^−1^ for HCQ and HCQ@CS NPs, respectively, calculated from the ratio of the intercept to the slope of the regression Equation (6), obtained from the plot of 1/Δ*A* vs 1/[*Q*] (Figure 9).
(6)$$\frac{1}{∆A}=\frac{1}{{(\varepsilon }_{b}-\varepsilon_{f})L_{T}}+\frac{1}{{(\varepsilon }_{b}-\varepsilon_{f})L_{T}K_{a}[M]}$$

Here in Equation (6), ε_b_ and ε_f_ are the extinction coefficients of bound and free ligand concentrations, *L*_T_ and M are the total ligand concentration and concentration of the macromolecule, respectively, Δ*A* is the change in the absorbance at a given wavelength, and *K*_b_ is the binding constant.

#### 2.3.5. Stopped-Flow Binding Experiments and Kinetic Measurements

Numerous parameters, including the strength of binding, the drug’s affinity for DNA, and the kinetic stability of DNA/protein–drug adducts, may contribute to antitumor efficacy because several drugs with reduced cellular toxicity have a slow DNA/protein dissociation rate [79]. Understanding the binding and reaction rates of frequently competing reactions, their dependence on cellular concentrations of participating molecules, and the regulation of these rates through posttranslational modifications or other mechanisms is necessary to comprehend cellular processes such as biochemical pathways and signaling networks. To do this, we dissect these systems into their fundamental components, which are usually always either unimolecular or bimolecular reactions that typically take place on time scales of less than a second, frequently less than a millisecond. For the study of such processes, stopped-flow mixing techniques, which often achieve mixing in less than 2 ms, are typically appropriate. Several excellent texts demonstrate how to use these methods to research enzyme processes [80,81,82]. We first concentrate on the calculation of the rate constants for processes that are common in protein–ligand interactions and are straightforward, reversible second-order reactions. Such reactions are typically studied using stopped-flow kinetics in pseudo-first-order settings with the concentration of one of the reagents at least tenfold higher than the other. In this study, HCQ and HCQ@CS NPs were selected to be employed in excess to maximize the ratio of signal change to the overall background signal.

BSA displays a distinctive ultraviolet (UV) absorption spectrum (Figure 10a) with a maximum at about 280 nm, primarily due to the aromatic amino acids tyrosine (Tyr) and tryptophan (Trp). It is common practice to determine the protein interaction using UV absorbance at 280 nm. The kinetic traces for HCQ@CS NPs were taken in 20 s at 260 nm, covering the full contact process. After the reaction with BSA during the first two seconds, the absorbance at 280 nm gradually dropped over the course of the following 20 s. The interaction between HCQ and BSA, in contrast, has a distinct pattern, with a steady increase in absorption over the period of 20 s.

The investigation of kinetic traces performed under pseudo-first-order conditions over the concentration range of HCQ and HCQ@CS NPs may be fitted to one and two kinetic phases (Figure 9 and Figure 10), evaluated using Equations (7) and (8), respectively:(7)$$A=a_{1}e^{-k_{obs}t}+A_{0}$$
(8)$$A=a_{1}e^{-k_{obs1}t_{1}}+a_{2}e^{-k_{obs2}t_{2}}+A_{0}$$

For HCQ@CS NPs, the stopped-flow records the two kinetic phases as a fast binding (rate constant *k*_obs1_) and a slow first-order isomerization process (rate constant *k*_obs2_). According to the equation *k*_obs_ = *k*_off_ + *k*_on_ [HCQ@CS NPs], both *k*_obs1_ and *k*_obs2_ increased linearly with HCQ@CS NPs concentration (Figure 10), with a slope equal to the second-order association constant, *k*_on_ [M^−1^s^−1^], and an intercept equal to the first-order dissociation constant, *k*_off_ [s^−1^]. The equilibrium association constants *K*_aff_ [*k*_on_/*k*_off_ M^−1^] and the equilibrium dissociation constants *K*_d_ [*k*_off_/*k*_on_ M] were calculated using both *k*_on_ and *k*_off_. According to the findings, which are shown in Table 4, BSA binds to HCQ@CS NPs reversibly with a second-order association constant of *k*_1_ = 175 ± 1 M^−1^s^−1^ and separates from the binary complex in the first phase with a first-order dissociation constant of *k*_-1_ = 7.9 ± 0.8 s^−1^. This indicates that the equilibrium dissociation constants, *K_d_*_1_, *k*_-1_*/k*_1_, is 4.5 × 10^−2^ M and that the binding affinity of BSA to HCQ@CS NPs, *K_a_*_1_, *k*_1_/*k*_-1_, of 22.2 M^−1^. In the second reaction step, a reversible reaction step with a coordination affinity of 10 M^−1^ for HCQ@CS NPs was found.

The stopped-flow records one kinetic phase for HCQ, and the observed rate constants were also found to be dependent on the HCQ concentration (Figure 11). A reversible reaction step with HCQ was reported, with *k*_1_ = 5.9 ± 0.4 M^−1^ s^−1^, *k*_−1_ = 3.5 ± 0.6 s^−1^, and a coordination affinity of 1.67 M^−1^. The equilibrium dissociation constant, *K*_d1_, is related to the variation in the free energy, *G*, of unbound BSA molecules in solution as compared to when they are bound together. The binding free energy change G_bind_ is provided by Equation (9) [83].
(9)$$∆G_{bind}=RT\cdot ln{(K}_{d})$$

While T and R stand for the universal gas constant and the absolute temperature, respectively. Equation (9) was used for calculating the *G*_bind_ values, which were 7.7 and 5.7 kJ mol^−1^ for the first and second binding phases of BSA to HCQ@CS NPs and 12.7 kJ mol^−1^ for HCQ. The negative sign for G indicates spontaneous binding to both HCQ and HCQ@CS NPs. From the individual equilibrium dissociation constants, *K*_d1_ and *K*_d2_, as well as the overall association constants, *K*_a_, Equation (10) demonstrates how to determine the overall equilibrium dissociation constant, *K*_d_.
(10)$$K_{d}=\frac{K_{d1}K_{d2}}{1+K_{d2}}$$

The information in Table 3 demonstrates that HCQ@CS NPs have a greater overall BSA binding affinity (245 M^−1^) compared to HCQ (167 M^−1^). Both HCQ@CS NPs and HCQ had *G*_bind_ values of −12.7 and −13.6 kJ mol^−1^, respectively, showing that the reaction is spontaneous for both compounds.

### 2.4. In Vitro Cytotoxicity Studies

Biocompatibility is an important concern when evaluating medicine delivery strategies for tumors. The anticancer activity of HCQ and HCQ@CS NPs was studied in vitro against a diseased cell line, the A549 lung cancer cell line, and the cytotoxic activity was assessed for one noncancerous human cell line, Wi38, using the 3-(4,5-dimethylthiazol)-2-diphenyltertrazolium bromide (MTT) assay in comparison to the standard anticancer drug *Doxo*. The cell viability was found to decrease as the concentration of HCQ and HCQ@CS NPs increased, showing that the anticancer potency of HCQ and HCQ@CS NPs was dose dependent. Figure 12 displays the outcomes of the drug cytotoxicity assay. The cytotoxic effects of HCQ and HCQ@CS NPs on A549 cells were tested at various doses, and the IC_50_ values of free drug and drug-loaded NPs for A549 cells were determined. The results showed that drug-loaded NPs were more cytotoxic to A549 cells than free HCQ and that both proved highly toxic to A549 cancer cells but less toxic to Wi38 normal cells. In addition to HCQ and HCQ@CS NPs, the activity of *Doxo*, a recognized clinically successful medication, was investigated in A549 and Wi38 cell lines under the same experimental conditions. In both cell lines evaluated, the drug-loaded NPs had much stronger anticancer activity than Doxo (Figure 12, Table 5). Figure 13 depicts the morphological alterations caused by HCQ, HCQ@CS NPs, and *Doxo* in A549 cancer cells and Wi38 normal cells. Figure 13 demonstrates that the control cells were uniformly shaped. The treated cells exhibited apoptosis-like shattered nuclei, brilliant staining, and condensed chromatin, and the modification in A549 malignancy was greater when the HCQ drug was incubated in CS nanoparticles. These data support the notion that HCQ@CS NPs are effective in inducing apoptosis in A549 cancer [84]. This may be due to increased functioning, such as cellular uptake. Proteins make up the second-largest portion of cell membranes, and some of them have the ability to mediate cellular uptake, also known as receptor-mediated uptake. When compared to HCQ (1.67 M^−1^), BSA protein had a higher affinity for HCQ@CS NPs (22.2 M^−1^), which may be a perfect chance to improve drug targeting and administration. [85,86]. A second reason may be the kinetic stability of the drug–DNA/protein complex, as Denny et al. [80] reported that the cytotoxicity of intercalating agents correlates better with the kinetic stability of the drug–DNA/protein complexes than with binding affinity. HCQ@CS NPs–BSA complex showed a higher stability over a long time compared to that of HCQ–BSA, as shown in Figure 14.

### 2.5. Antibacterial Activity

Because chemotherapy makes cancer patients immunocompromised and prone to infections, cancer patients are routinely given a multi-drug cocktail of antimicrobial drugs in addition to chemotherapeutic medications to prevent or eliminate different infections [73]. As a result, the antibacterial properties of HCQ@CS NPs and HCQ were also tested in vitro using four bacteria strains: Two Gram-positive (*E. faecalis* and *S. aureus*) and two Gram-negative (*P. aeruginosa* and *E. coli*) using the DTM technique. The bacterial strains were grown in the presence of HCQ@CS NPs and HCQ (concentrations ranging from 0 to 1000 μg/mL) to determine the inhibition zone and compared with a standard antibiotic (chloramphenicol). HCQ@CS NPs and HCQ are active against both Gram-(+) and Gram-(−) bacteria compared to chloramphenicol under the same experimental condition. As can be seen in Figure 15, HCQ@CS NPs afforded higher activity against all tested bacteria than free chloramphenicol. The antibacterial function is a complicated process that varies between G− and G+ bacteria due to the differences in cell wall and cell membrane structure. HCQ@CS NPs and HCQ exhibit greater antibacterial activity against G− bacteria than G+ bacteria, probably because of the G− bacterial cell membrane. The G− bacteria cell membrane has lipopolysaccharide (LPS) that contains phosphate and pyrophosphate anionic groups. These groups provide more negative charges to the cell surface compared to G+ bacteria, which contain peptidoglycan and teichoic acid instead. This may explain why the loss of intracellular contents observed with HCQ@CS NPs in G− is greater than that in G+ bacteria [87]. The reason for the improved antibacterial activity of HCQ@CS NPs compared to free HCQ may be related to the particle size. The size range of HCQ@CS NPs was 159.3 ± 7.1 nm, which is smaller than that of HCQ (255.7 ± 18.45 nm). The smaller size of HCQ@CS NPs increases cellular absorption because the antibacterial activity depends heavily on the particle size, as stated by Zhang et al. [88]. Another reason for the high antibacterial activity of HCQ@CS NPs compared to free HCQ may be the zeta potential, as reported by Dey et al. [89]. The zeta potential of HCQ@CS NPs (+46.6 ± 0.8 mV) and that of HCQ (−16.4 mV) support the higher activity of HCQ@CS NPs compared to the free HCQ and also support the higher activity of both against G− bacteria than G+ bacteria.

### 2.6. Molecular Docking of BSA with HCQ

The Molecular Operating Environment (MOE) software [90] was used to investigate the molecular docking between drugs or ligands and a variety of biological targets, such as BSA. Docking simulation for the HCQ–BSA complex allows the estimation of energy. The best leads superimposed core cavity interaction was visualized (Figure 16). Following molecular docking, the best position of HCQ was in the protein structure, with a binding affinity of DG −7.0 kcal/ mol (−29.4 kJ/mol). The ligand, HCQ, is positioned in the BSA binding site, surrounded by Thr 578, Gln 579, Gln 525, Tyr 400, and Asn 404. It binds by greasy forces (backbone donor) with residues of Leu574, Leu531, Leu528, Val575, Val546, Val551, Phe501, Phe506, Phe550, Ale527, and Met547. The termination of HCQ (CH_2_OH) is considered ligand exposure. HCQ also has two interacting groups with OG1 and Thr578 as hydrogen donors, with a bond distance of 2.8 Ǻ.

## 3. Conclusions

In conclusion, a tailored HCQ@CS NPs nanocomposite was synthesized with the goal of producing a novel nanocomposite with better properties and bioavailability. FTIR, SEM, DLS, and zeta potential methods were used to characterize the HCQ@CS NPs nanocomposite. HCQ@CS NPs have a smooth surface and are roughly shaped like roadways, with an average size of 159.3 ± 7.1 nm, a PdI of 0.224 ± 0.101, and a zeta potential of +46.6 ± 0.8 mV. The binding mechanism and affinity of the interaction between HCQ and HCQ@CS NPs and BSA were investigated using stopped-flow, various spectroscopic approaches, and molecular docking studies to aid in the creation of pharmacological systems for application in cancer therapy. HCQ and HCQ@CS NPs binding with BSA is driven by a ground-state complex formation that may be accompanied by a non-radiative energy transfer process, and binding constants at a single site indicated that HCQ@CS NPs-HCQ was more stable than HCQ–BSA. This was demonstrated by UV–vis and fluorescence measurements. Stopped-flow investigations showed that HCQ@CS NPs bind through two reversible steps: A fast second-order binding, followed by a slow first-order isomerization reaction via a static quenching mechanism. For the first and second steps of HCQ@CS NPs, the detailed binding parameters were established. The total binding constants for HCQ@CS NPs (*K*_a_ = 244.5 M^−1^, *K*_d_ = 0.4 × 10^−2^ M^−1^, Δ*G*^0^ = −13.6 kJ mol^−1^). In the case of HCQ, however, just one reversible reaction step was detected with a *K*_a_ = 176.0 M^−1^, *K*_d_ = 0.6 × 10^−2^ M^−1^, and Δ*G*^0^ = −12.6 kJ mol^−1^. These findings showed that, in addition to increasing BSA affinity, the nanoformulation HCQ@CS NPS changes the binding process and may open new routes for interaction. Docking experiments verified the development of the HCQ–BSA complex, with HCQ binding to site I on the BSA structure, primarily with the amino acids Thr 578, Gln 579, Gln 525, Tyr 400, and Asn 404. Furthermore, the nanoformulation HCQ@CS NPS not only demonstrated increased cytotoxicity against the A549 lung cancer cell line (IC_50_ = 28.57 ± 1.72 μg/mL) compared to HCQ (102.21 ± 0.67 μg/mL) but also demonstrated higher antibacterial activity against G bacteria than G+ bacteria when compared to HCQ and chloramphenicol. The nanoformulation developed in this study may offer a viable therapy option for A549 lung cancer.

## 4. Materials and Methods

### 4.1. Materials

Sigma-Aldrich Chemicals, Germany provided the low-molecular-weight CS (2.6 kDa MW and 80–85% DD), TPP, acetic acid, and BSA, while Pharmaceutical, india, provided the HCQ.

### 4.2. Preparation of HCQ@CS NPs

Low molecular weight CS 0.6 g was dissolved at 25 °C by adding acetic acid (30 mL, 2 M) progressively and stirring at 298 K to obtain a solution containing CS NPs. The pH of the solution was brought down to 5.5 by adding a 2 M NaOH solution. The solution was filtered using a 0.45 m cellulose acetate filter to get rid of any remaining CS. Over the course of an hour, HCQ (0.358 g, 0.0011 mol) was dropped into the CS solution while being stirred. By dissolving 0.2 weight percent of TPP in double-distilled water and filtering it through 0.25-micron cellulose acetate, a TPP solution was formed. The HCQ–chitosan solution was then given 2 mL of TPP, which was added dropwise by burette at a rate of 0.2 mL/min while being agitated at room temperature. While centrifuging the remaining samples for 20 min, some of the processed samples were utilized to measure the nanoparticle size. The samples were then centrifuged once more to get rid of the unreacted gradients after being rinsed with water. The nanoparticles were then analyzed after being dried by air at room temperature. According to the nanoparticle characterization technique, the drug-loaded NPs were characterized using FT-IR, DLS, and zeta potential. The elemental analysis of HCQ@CS NPS was determined using a Heraeus Carlo Ebra 1108 Elemental Analyzer: C, 37.3; N, 6.94%.

### 4.3. Characterization

To evaluate the linking of functional groups in CS, CS NPs, and drugs to the nanoparticles, an FT-IR spectrometer (Perkin Elmer, Germany) was used, and the spectra were collected over a range of 400–4000 cm^−1^. To examine the effects of the mode of production procedure and other process parameters, the mean droplet size and size distribution in the form of PDI of the diluted (1:100) NE were determined by the DLS technique utilizing a zetasizer (Malvern Instruments, 1000 HS, Malvern, UK) in a tris-buffer solution (tris(hydroxymethyl)aminomethane). DLS with noninvasive back scattering (DLS-NIBS) was used to determine the hydrodynamic diameter of the samples, assuming a spherical shape, with a laser wavelength of 632 nm and a fixed scattering angle of 173°. The zeta potential (ζ) was determined in a tris-buffer solution using a combination of mixed laser Doppler electrophoresis and phase analysis light scattering (M3-PALS). A Malvern Zetasizer Nano, Model ZS 3600 (Malvern Instruments, Worcestershire, UK) was used for both measurements. The measurements were taken three times. The particle size and zeta potential of the dispersion were measured without the isolation of nanoparticles. The morphology investigation of the prepared sample was performed using SEM (Seron Technology, South Korea). A UV–vis spectrophotometer (Model Perkin-Elmer Lambda) was used to assess optical characteristics. 

### 4.4. BSA Binding Study Procedures

The appropriate solid BSA was dissolved in distilled water to produce a BSA (10^−4^ M) solution. In methanol, 10^−3^ M stock solutions of HCQ and HCQ@CS NPs were prepared. UV–vis measurements were carried out between 240 and 600 nm using a UV-1800 Shimadzu spectrophotometer. When measuring the absorbance values of BSA in the absence and presence of samples, the concentrations of HCQ and HCQ@CS NPs were changed while the concentration of BSA (0.1 mM) remained constant. Equation (6) was used to calculate the binding constant (*K*_b_). Fluorescence measurements were taken in a quartz cell with a 1 cm path length using a Shimadzu spectrofluorometer model RF-5301 connected to a PC. The fluorescence spectra were obtained between 287 and 500 nm at k_em_ and at k_exc_ = 280 nm. According to earlier literature findings, the binding constant (K) was calculated using the intensity at 338 nm (tryptophan) (Equations (1)–(5)).

### 4.5. Stopped-Flow Fluorescence Kinetic Studies

The kinetic measurements were carried out with the Applied KinetAsyst SF-61DX2 stopped-flow device (HI-Tech Scientific) and a Peltier thermostat. The BSA concentration was 10^−4^ M, and various concentrations of the HCQ and HCQ@CS NP solutions were utilized. First, the two different syringes of the kinetic accessory were filled with BSA and HCQ or HCQ@CS NP solutions. Then, equal amounts of both solutions were simultaneously injected into the sample chamber, and this process was repeated for each run. The BSA’s absorption spectra were continuously monitored before (t = 0 s) and after the injection of HCQ or HCQ@CS NPs. When the instrument’s dead time was measured for a 1:1 mixture, it was discovered to be 2 ms. In control studies, BSA solution was mixed with buffer solutions containing either no HCQ or HCQ@CS NPs. The decomposition of the BSA was excluded because the absorption signal did not change over the course of the control experiment. Due to the complexity of the kinetic trace, which revealed that there was only one step present in the case of HCQ whereas two steps were recorded for HCQ@CS NPs, we assumed a superposition of exponential terms to characterize the process.

### 4.6. Cell Cytotoxicity Assay

To test the cytotoxicity of HCQ and HCQ@CS NPs, A549 cells were seeded onto 96-well plates at a density of 1 × 10^−5^/mL (100 L per well), and then HCQ and HCQ@CS NPs were added at final concentrations of 50, 125, 250, 500, and 1000 g/L (all in triplicate). Untreated cells served as a positive control for viability, while 2% saponin (Sigma-Aldrich, St. Louis, MO, USA) served as a positive control for cell lysis. Plates were incubated for 24 h under normal conditions. The ability of live cells to decrease MTT (Cell Proliferation and Cytotoxicity Assay, R&D Systems, Minneapolis, MN, USA) to formazan crystals was used to assess cell viability using the MTT colorimetric approach. Following the 24 h incubation period, 50 L of MTT solution was applied to each well. After four hours of incubation (37 °C, 5% CO_2_), the plate was centrifuged (400× *g*/10 min), supernatants were removed, and the intracellularly stored formazan was solubilized for four hours at room temperature with 200 L of the dissolving solution. The optical density was then measured using a plate-reading spectrophotometer Victor 2 (Wallac, Turku, Finland) at a reference wavelength of 620 nm. The values of the half-maximal inhibitory concentration (IC_50_) were then computed.

### 4.7. Antibacterial Activity

The in vitro antibacterial activity of HCQ and HCQ@CS NPs against Gram-positive (*S. aureus* and *E. faecalis*) and Gram-negative (*E. coli* and *P. aeruginosa*) bacteria was examined using the DTM method [91]. The examined HCQ and HCQ@CS NPs were diluted into a range of concentrations of 5, 6, 7, 7.5, 9.0, 9.5, 9.8, 10, and 15 g/mL in test tubes filled with sterile nutritional broth. In test tubes containing 1 mL of the various concentrations of the free drug and its CS NPs in nutritive broth, inoculated *E. coli* culture, 0.5 McFarland standards. After 18 to 24 h at 37 °C, the tubes were examined for growth or turbidity. Each test tube that did not exhibit growth had a loopful of broth put into a plate of nutritional agar. The test tube cultures were then fed equal volumes of sterile nutrient broth and cultured at 37 °C for another 24 h. The tubes and agar plates were then examined visually for growth and turbidity (CLSI, 2012). Instead of doubling the nutritional broth dilution, the DTM was repeated with a greater dilution (1:10 dilution, or 9 vol. of nutrient broth to 1 vol. of broth culture). The same experiments were carried out three times. Chloramphenicol was utilized as a control drug.

### 4.8. Computational Studies

All theoretical calculations were carried out using Gaussian 09 W [53] and density function theory (DFT) [54,55]. The Becke 3-Parameter, Lee, Yang, and Parr (B3LYP) [56,57] with a 6–31 G(d) basis set was used to optimize the structure of HCQ, the monomer of chitosan (M), and the dimer of chitosan (MM). In the gas phase, all computations were fully optimized. Frequency calculations were performed at the same theoretical level for each optimized shape to ensure that they are all minima or transition states on the system’s potential energy surface. The ChemCraft program was used to calculate the distribution of the highest occupied molecular orbital (HOMO), lowest unoccupied molecular orbital (LUMO), and frontier molecular orbitals (FMO) in HCQ, M, and MM [58]. The Gaussian View package [92] was used to generate the molecular electrostatic potential (MEP).

### 4.9. Molecular Docking

MOE software [90] was utilized for molecular docking studies between drugs or ligands and a variety of biological targets, such as BSA. MOE is also a drug discovery software platform that combines modeling, simulation, and visualization, as well as methodological development [93]. HCQ was optimized by minimizing energies using parameters and the best model was chosen for the docking study. The MOE docking program with default parameters was used to attach the specified ligand to the receptor protein and provide the correct substrate shape.

## Figures and Tables

**Figure 1 ijms-24-14103-f001:**
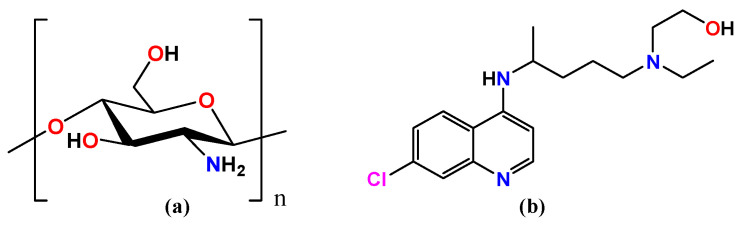
Structural formula of (**a**) chitosan, and (**b**) hydroxychloroquine.

**Figure 2 ijms-24-14103-f002:**
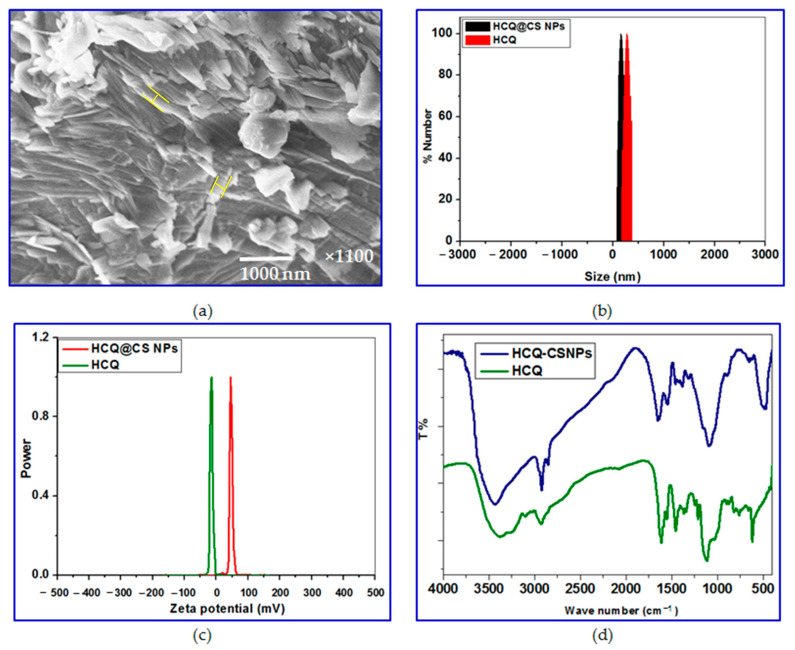
(**a**) SEM images of HCQ@CS NPs under 500 nm field of view; shape-like roads are highlighted; (**b**) the particle size distribution of HCQ and HCQ@CS NPs detected by Zetasizer Nano ZS; (**c**) zeta potential of HCQ, and HCQ@CS NPs; (**d**) the chemical structure of HCQ, and HCQ@CS NPs are analyzed by FT-IR.

**Figure 3 ijms-24-14103-f003:**
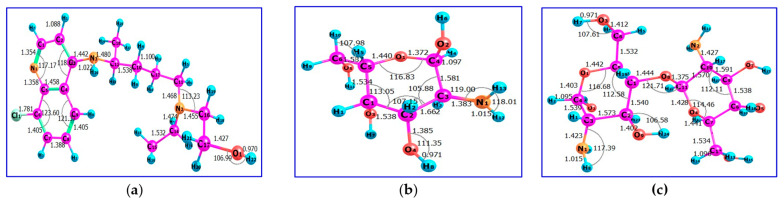
The optimized geometries of (**a**) HCQ, (**b**) M and (**c**) MM.

**Figure 4 ijms-24-14103-f004:**
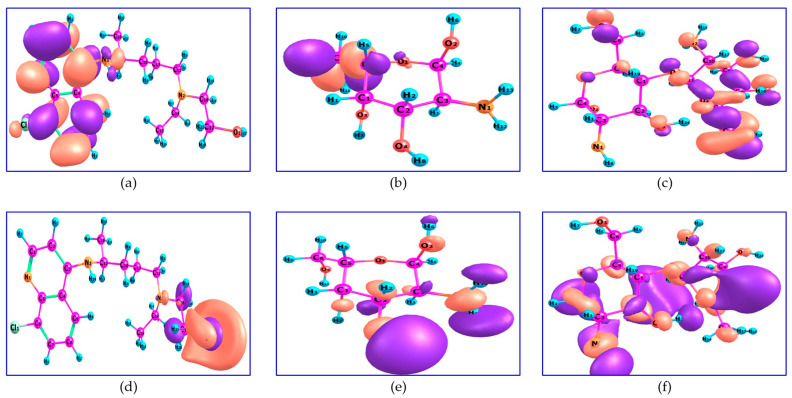
HOMO of (**a**) HCQ; (**b**) M and (**c**) MM and LUMO of (**d**) HCQ; (**e**) M and (**f**) MM.

**Figure 5 ijms-24-14103-f005:**
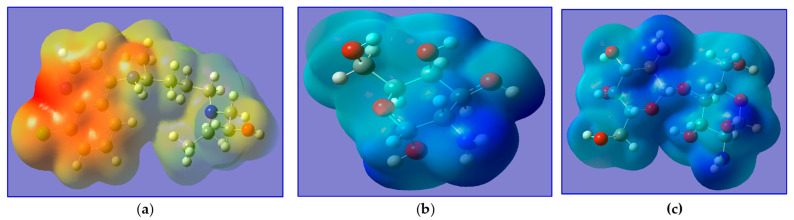
MEP of (**a**) HCQ, (**b**) M, and (**c**) MM.

**Figure 6 ijms-24-14103-f006:**
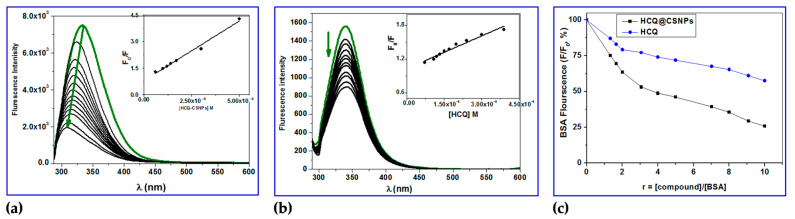
Fluorescence quenching spectra of BSA in the absence (green) and presence (black) of various concentration of (**a**) HCQ@CSNPs and (**b**) HCQ at T = 298 K, λ_exc_ = 278 nm, λ_em_ = 343 nm, pH = 7.2 and [BSA] = 4 × 10^−4^ M; green arrows show the quenching; insets show Stern–Volmer plots; (**c**) Plot of relative BSA fluorescence intensity at λ_em_ = 343 nm (F/F_0_, %) vs. r (r = [compound]/[BSA]) for HCQ@CSNPs and HCQ at T = 298 K, λ_exc_ = 278 nm, λ_em_ = 343 nm, pH = 7.2 and [BSA] = 4 × 10^−4^ M. Arrows showed the quenching with or without wavelength shift.

**Figure 7 ijms-24-14103-f007:**
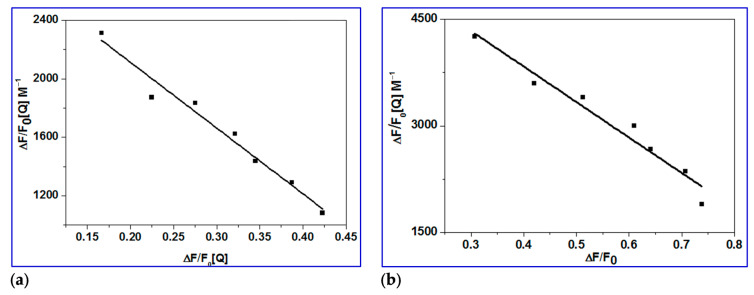
Scatchard plots of the reaction of BSA with (**a**) HCQ and (**b**) HCQ@CSNPs.

**Figure 8 ijms-24-14103-f008:**
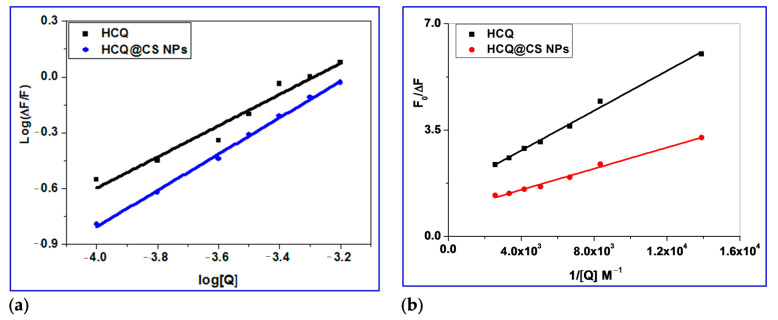
(**a**) Double logarithmic plot of BSA (0.4 mM) fluorescence quenching in presence HCQ@CSNPs and HCQ at different concentrations; (**b**) Modified Stern–Volmer plot; the accessibility of BSA to HCQ@CSNPs and HCQ.

**Figure 9 ijms-24-14103-f009:**
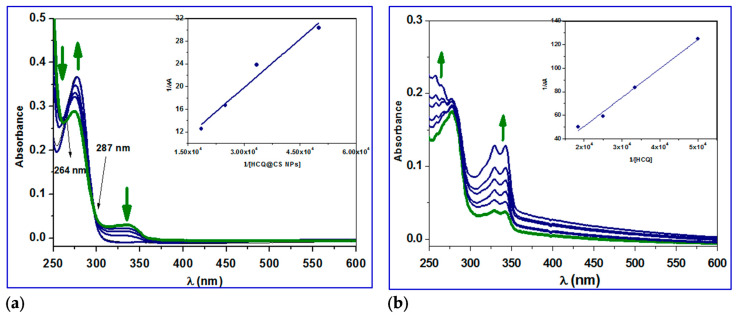
UV–vis spectral changes of the reaction of BSA in the absence (green line) and presence (blue lines) of different concentrations of (**a**) HCQ@CS NPs and (**b**) HCQ in methanol at 296 K. Arrows showed the hypochromic and hyperchromic effects.

**Figure 10 ijms-24-14103-f010:**
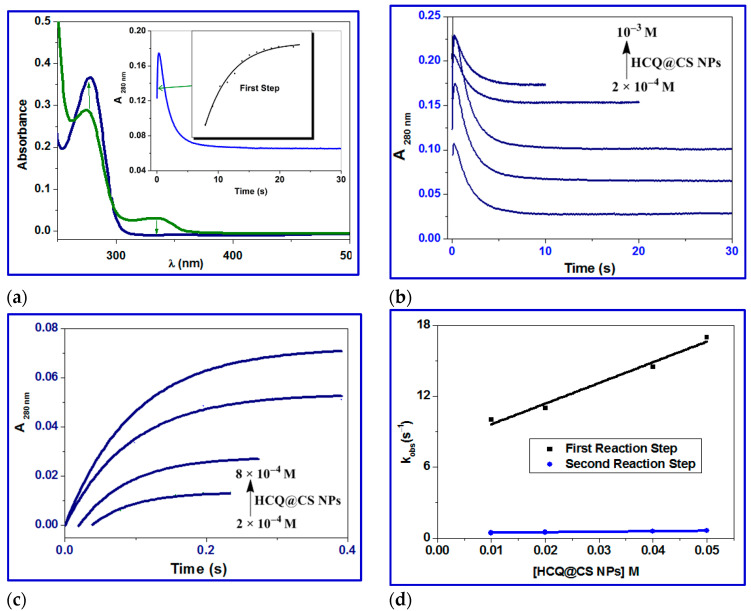
(**a**) UV–vis spectral changes of binding of BSA in the absence (green line) and presence (blue line) of HCQ@CS NPs; green arrows show the hyperchromism and hypochromism; inset showing the kinetic trace at 280 nm containing two reaction steps; (**b**) kinetic traces of different concentrations of HCQ@CS NPs (2 × 10^−4^, 4 × 10^−4^, 6 × 10^−4^, 8 × 10^−4^, 10^−3^ M) with BSA; (**c**) fast step of the reaction of different concentrations of HCQ@CS NPs (2 × 10^−4^, 4 × 10^−4^, 6 × 10^−4^, 8 × 10^−4^ M) with BSA; (**d**) plots of *k*_obs_ versus concentration of the first and second reaction steps of BSA with HCQ@CSNPs in methanol at 296 K.

**Figure 11 ijms-24-14103-f011:**
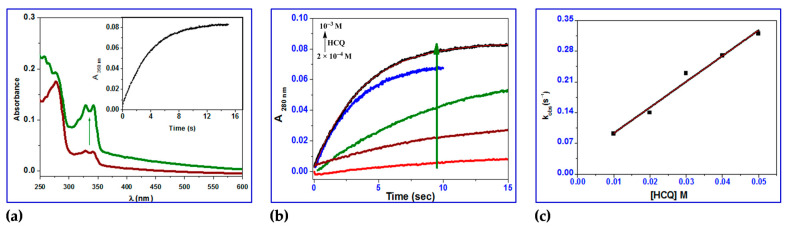
(**a**) UV–vis spectral changes of binding of BSA in the absence (brown line) and presence (green line) of HCQ; inset showing the kinetic trace at 280 nm containing two reaction steps; (**b**) kinetic traces of different concentrations of HCQ (2 × 10^−4^, 4 × 10^−4^, 6 × 10^−4^, 8 × 10^−4^, 10^−3^ M) with BSA; (**c**) plots of *k*_obs_ versus concentration of the first and second reaction steps of BSA with HCQ in methanol at 296 K.

**Figure 12 ijms-24-14103-f012:**
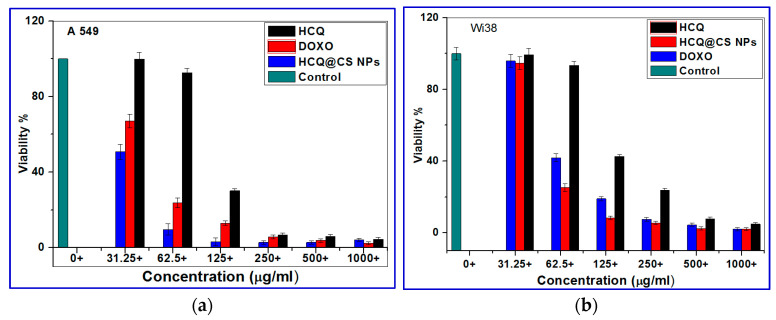
Cell viability results from the MTT assay for (**a**) A549 cancer cells and (**b**) Wi38 normal cells with HCQ, HCQ@CS NPs and *Doxo* for 24 h.

**Figure 13 ijms-24-14103-f013:**
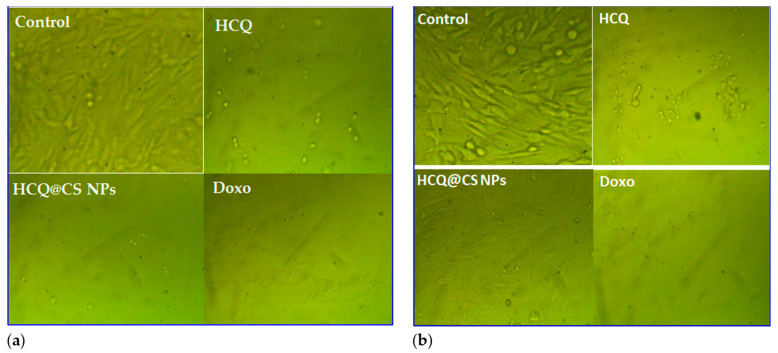
Morphological changes in (**a**) Wi38 normal cells and (**b**) A549 cancer cells following 24 h of treatment with control, HCQ, HCQ@CS NPs, and *Doxo* at 1000 μg/mL concentrations. Scale bar: 25 μm.

**Figure 14 ijms-24-14103-f014:**
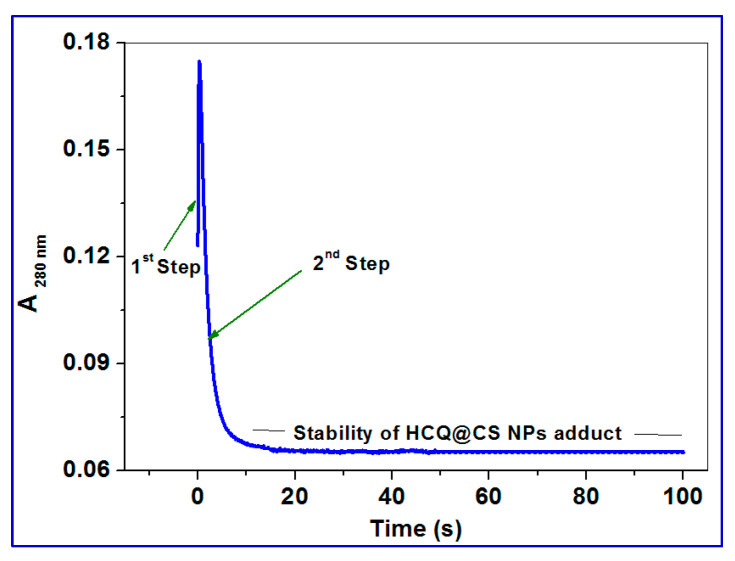
Kinetic trace of showing the stability of HCQ@CS NPs–BSA adduct over longer time.

**Figure 15 ijms-24-14103-f015:**
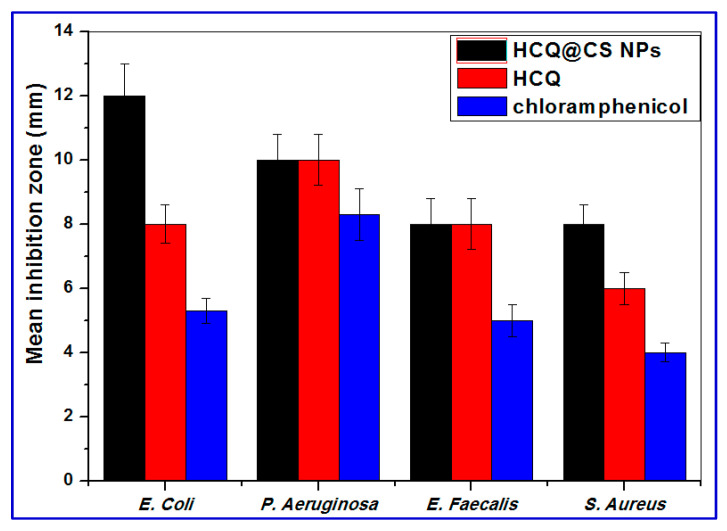
Mean inhibition zone of the HCQ free drug and its loaded nanoparticles against Gram-negative (*E. coli* and *P. aeruginosa)* and Gram-positive (*S. aureus* and *E. faecalis)* bacteria.

**Figure 16 ijms-24-14103-f016:**
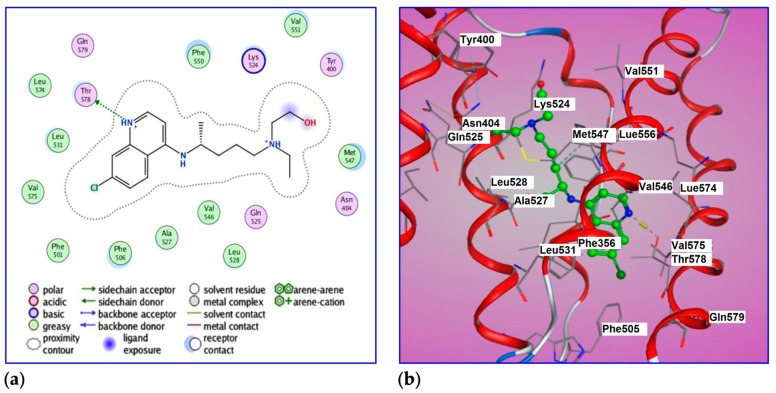
(**a**) Two-dimensional (2D) and (**b**) 3D diagrams of ligand showing its interaction with crystal structure of BSA (4F5S) binding site.

**Table 1 ijms-24-14103-t001:** Mean diameter and zeta potential of HCQ and HCQ@CS NPs.

	Size (nm)	PDI	Zeta Potential (mV)
HCQ	255.7 ± 18.5	0.224 ± 0.083	−16.4 ± 0.4
HCQ@CS NPs	159.3 ± 7.1	0.337 ± 0.101	+46.6 ± 0.8

**Table 2 ijms-24-14103-t002:** Quantum parameters for HCQ; M and MM.

Parameter	*E*_HOMO_ (eV)	*E*_LUMO_ (eV)	IP	EA	*E*_LUMO_–*E*_HOMO_ (eV)	Χ (eV)	Pi (eV)	η (eV)	σ (eV^−1^)	S (eV^−1^)	ω (eV)
HCQ	+1.46	+3.42	−1.46	−3.42	1.96	−2.44	2.44	0.982	1.018	0.509	3.037
M	−10.27	−3.07	+10.27	+3.07	7.2	6.67	−6.67	3.601	0.277	0.138	6.177
MM	−10.15	−1.78	+10.15	+1.78	8.36	5.97	−5.97	4.183	0.239	0.119	4.261

**Table 3 ijms-24-14103-t003:** Quenching, binding, and thermodynamic parameters of BSA–HCQ and BSA–HCQ@CS NPs complexes.

	*K*_SV_ [M^−1^ × 10^4^]	*K*_q_ [M^−1^s^−1^ × 10^12^]	*n*	*f* _a_	*K* [M^−1^ × 10^4^]	*K_a_* [M^−1^ × 10^4^]	Δ*G* [kJ Mol^−1^]	*K*_b_ [M^−1^ × 10^4^]
HCQ	0.20	0.20	0.8	1.10	0.41	0.15	−7.8	0.10
HCQ@CS NPs	0.70	0.70	1.0	1.21	0.62	0.80	−9.7	0.55

**Table 4 ijms-24-14103-t004:** Rate constants, affinities, and dissociation constants recorded during the reaction of BSA with HCQ and its nanoparticles, HCQ@CSNPs at 298 K.

First Step						Second Step					Overall Reaction		
	*k*_1_ [M^−1^s^−1^]	*k*_-1_ [s^−1^]	*K*_a1_ [M^−1^]	*K*_d1_ [M]	Δ*G*_1_ [kJ mol^−1^]	*K*_2_ [M^−1^s^−1^]	*k*_-2_ [s^−1^]	*K*_a2_ [M^−1^]	*K*_d2_ [M]	Δ*G* [kJ mol^−1^]	*K*_d_ [10^−2^M]	*K*_a_ [M^−1^]	Δ*G* [kJmol^−1^]
HCQ@CS NPs	175 ± 10	7.9 ± 0.8	22.2	0.045	−7.7	4.1 ± 0.2	0.4 ± 0.0	10	0.1	−5.7	0.4	244.5	−13.6
HCQ	5.9± 0.4	3.5 ± 0.6	1.67	0.6	−12.7	Only one step has been detected							

**Table 5 ijms-24-14103-t005:** IC_50_ values obtained for HCQ, HCQ@CS NPs, and *Doxo* from the MTT assay using A549 and Wi38 cells.

	IC_50_ Values ^1^ (μg/mL)	
	Wi38	A549
HCQ	115 ± 1.4	102.21 ± 0.67
HCQ@CS NPs	66.37 ± 0.51	28.57 ± 1.72
Doxo	57.81 ± 0.79	50.68 ± 3.52

^1^ IC_50_ values are the mean concentrations of drugs to inhibit 50% of cancer cells (in μM). An average of three replicates is taken and presented as mean ± SD.

## Data Availability

Not applicable.

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
