# Peer review of "Hydroxychloroquine-Loaded Chitosan Nanoparticles Induce Anticancer Activity in A549 Lung Cancer Cells: Design, BSA Binding, Molecular Docking, Mechanistic, and Biological Evaluation"

_ijms, 2023, doi:10.3390/ijms241814103_

Round 1
Reviewer 1 Report
The novelty and quality of presentation of this paper is not suitable for IJMS. In addition, the interpretation/discussion is sometimes more than what can truly be said from the obtained results, which is causing leaps in logic from point to point. Careful revision of the experimental design as well as additional data where appropriate are inevitable to provide sufficient evidence behind the statements that the authors make. I just give a few examples.
1. "Hydroxychloroquine-Based Chitosan Nanoparticles" is not correct. Hydroxychloroquine is loaded into NPs. Please modify the similar description in the entire manuscript.
2. "By taking advantage of the electrostatic interactions between the positively charged amino groups in CS and the hydroxyl group of HCQ". Are you sure that the encapsulation is from electrostatic interactions?
3. TME images are very unclear. The readers can not distinguish NPs.
4. The title is too long.
In fact, a lot of issues besides....
Author Response
Reviewer 1: The novelty and quality of presentation of this paper is not suitable for IJMS. In addition, the interpretation/discussion is sometimes more than what can truly be said from the obtained results, which is causing leaps in logic from point to point. Careful revision of the experimental design as well as additional data where appropriate are inevitable to provide sufficient evidence behind the statements that the authors make. I will just give a few examples.
Author response: the authors are very much thankful for the reviewers for thorough reviewing the manuscript.
- "Hydroxychloroquine-Based Chitosan Nanoparticles" is not correct. Hydroxychloroquine is loaded into NPs. Please modify the similar description in the entire manuscript.
Author response: That is true and was corrected as suggested by the reviewer.
- "By taking advantage of the electrostatic interactions between the positively charged amino groups in CS and the hydroxyl group of HCQ". Are you sure that the encapsulation is from electrostatic interactions?
Author response: We think that the positively charged amino groups may have electrostatic interaction with hydroxyl group from the HCQ and the positively charged amino groups may be the reason for the high positive value of the zeta potential as well.
- TME images are very unclear. The readers cannot distinguish NPs.
Author response: That is true, and we added a better one.
- The title is too long.
Author response: That is true and now it is shorter. "Hydroxychloroquine-Loaded Chitosan Nanoparticles Induce Anticancer Activity in A549 Lung Cancer Cells: Design, BSA Binding, Molecular Docking, Mechanistic, and Biological Evaluation"
In fact, there are a lot of issues besides.
Reviewer 2 Report
In this work the authors investigate the possibility of obtaining conjugates of chitosan with hydroxychloroquine with antimicrobial and antitumor activity. This work is topical and can arouse a certain interest of readers, but in this form the manuscript contains quite a lot of remarks and requires mandatory revision with additional experiments and editing of the English language.
1) The first sentence of the manuscript contains tautology and does not make sense.
2) There is no discussion of the causes of lung cancer development. The causes of DNA damage should be summarized, as well as the reasons for the lack of response of the body to the mutated cells formed under appropriate pathological conditions.
3) Reference 9 discusses the use of chitosan as a carrier of drugs in ophthalmology when applied topically, whereas the antitumor effect of the materials under development is possible with their systemic action. More adequate examples should be given.
4) All abbreviations should be deciphered at their first mention and used further in the text.
5) In section 2.1. the authors state that the obtained CS-HSQNPs materials have a Z-potential of +46.6, whereas below they write that the antitumor effect is due to the absorption of negatively charged drug particles onto the cell surface. Should be corrected.
6) The authors did not present the band corresponding to the N-H bond vibrations in the spectra of chitosan.
7) The data of IR spectroscopy cannot be used to state the chemical interaction of HSQ with CS because the conditions of IR spectra acquisition are not quantitative in this case.
8) The paper absolutely lacks data on the molecular structure of the used research objects. The authors should introduce information on the structural formula of CS, its molecular weight characteristics and solubility in section 2.1.
9) The authors should provide in the introduction a justification for the choice of chitosan as a carrier for HSQ, since its choice is not obvious.
10) The authors should provide in the experimental part the elemental analysis and 13C NMR spectroscopy data of HSQ and CS.
11) In section 4.3, the authors again use the full names of substances instead of the abbreviations introduced earlier.
12) In which section of the manuscript are the XRD analysis data presented?
13) The authors should provide data to determine the hydrodynamic size of nanoparticles and their Z-potential under biomimetic conditions (pH 7.35, saline solution).
14) Writing of bacterial cultures should be italicized.
15) Figures 11 and 12 must necessarily contain error bars with an assessment of the validity of the manuscript presented in the text.
The work contains many grammatical errors and needs to be checked by a native English speaker.
Author Response
Reviewer 2: In this work the authors investigate the possibility of obtaining conjugates of chitosan with hydroxychloroquine with antimicrobial and antitumor activity. This work is topical and can arouse a certain interest of readers, but in this form the manuscript contains quite a lot of remarks and requires mandatory revision with additional experiments and editing of the English language.
Author response: the authors are very much thankful for the reviewers for thorough reviewing the manuscript.
1) The first sentence of the manuscript contains tautology and does not make sense.
2) There is no discussion of the causes of lung cancer development. The causes of DNA damage should be summarized, as well as the reasons for the lack of response of the body to the mutated cells formed under appropriate pathological conditions.
Author response for comments 1 and 2: This part has been modified and the mechanism was reported with related references as suggested by the reviewer
3) Reference 9 discusses the use of chitosan as a carrier of drugs in ophthalmology when applied topically, whereas the antitumor effect of the materials under development is possible with their systemic action. More adequate examples should be given.
Author response: More examples are added as suggested by the reviewer
4) All abbreviations should be deciphered at their first mention and used further in the text.
Author response: OK, have been revised carefully and corrected
5) In section 2.1. The authors state that the obtained CS-HSQNPs materials have a Z-potential of +46.6, whereas below they write that the antitumor effect is due to the absorption of negatively charged drug particles onto the cell surface. Should be corrected.
Author response: Sorry for that; was corrected.
6) The authors did not present the band corresponding to the N-H bond vibrations in the spectra of chitosan.
Author response: unfortunately, the O-H at 3514 cm-1 is broad and interferes with the NH band but another characteristic band at 1596 cm-1 has been found and described.
7) The data of IR spectroscopy cannot be used to state the chemical interaction of HSQ with CS because the conditions of IR spectra acquisition are not quantitative in this case.
Author response: Off course IR spectra are not quantitative and for this reason other tools are also reported.
8) The paper absolutely lacks data on the molecular structure of the research objects used. The authors should introduce information on the structural formula of CS, its molecular weight characteristics and solubility in section 2.1.
Author response: Thank you for this comment and we have added structures, data and other information in the text.
9) The authors should provide in the introduction a justification for the choice of Chitosan as a carrier for HSQ, since its choice is not obvious.
Author response: Yes, we agree with the reviewer and the text was corrected as suggested by the reviewer.
10) The authors should provide in the experimental part the elemental analysis and 13C NMR spectroscopy data of HSQ and CS.
Author response: Yes, but due to the bad solubility, we cannot get good 13C NMR data.
11) In section 4.3, the authors again use the full names of substances instead of the abbreviations introduced earlier.
Author response: Yes, corrected as suggested by the reviewer.
12) In which section of the manuscript is the XRD analysis data presented?
Author response: It was a mistake and was removed.
13) The authors should provide data to determine the hydrodynamic size of nanoparticles and their Z-potential under biomimetic conditions (pH 7.35, saline solution).
Author response: The comment is nice, and work is now undergoing in more details at different pH media including the biomimetic conditions.
14) Writing of bacterial cultures should be italicized.
Author response: was corrected.
15) Figures 11 and 12 must necessarily contain error bars with an assessment of the validity of the manuscript presented in the text.
Author response: were corrected and error bars were added.
Reviewer 3 Report
The authors synthesized a tailored HCQ@CS NPs nanocomposite, which have increasing BSA affinity. The nano-formulation not only enhanced cytotoxicity against the A549 but also exhibited higher anti-bacterial activity especially against Gram-negative bacteria. This HCQ@CS NPs developed in this study may offer a viable therapy option for A549 lung cancer. The manuscript is certainly attractive but a few minor concerns need to be addressed carefully before acceptance.
1. Author claimed that the HCQ@CS NPs was roughly shaped like roadways, can author explain why the specific structure formed?
2. The characterization of the nanoparticles is insufficient. For example, stability test of HCQ@CS NPs in serum were missing and should be added. The drug release study should also be further analyzed.
3. Since the nanoparticle is prone to be covered with various proteins in circulation, why author only put emphasis on BSA binding study without detecting any other possible protein?
4. The positive-charged HCQ@CS NPs may have serious safety issue when applied in vivo, how to resolve this potential risk?
The title was too redundant to understand. Please re-word it and make it concise.
Author Response
Reviewer 3: The authors synthesized a tailored HCQ@CS NPs nanocomposite, which has increasing BSA affinity. The nano-formulation not only enhanced cytotoxicity against the A549 but also exhibited higher anti-bacterial activity especially against Gram-negative bacteria. This HCQ@CS NPs developed in this study may offer a viable therapy option for A549 lung cancer. The manuscript is certainly attractive, but a few minor concerns need to be addressed carefully before acceptance.
Author response: the authors are very much thankful for the reviewers for thorough reviewing the manuscript.
- The authors claimed that the HCQ@CS NPs was roughly shaped like roadways, can author explain why the specific structure formed?
Author response: HCQ@CS NPs were prepared via a combination of HCQ and cross-linking process of chitosan in the presence of water and TPP. The road-like shape might have been due to stronger hydrogen bonds between HCQ and chitosan particles, enhanced by TPP presence.
- The characterization of the nanoparticles is insufficient. For example, stability test of HCQ@CS NPs in serum were missing and should be added. The drug release study should also be further analyzed.
Author response: Work is now undergoing in more detail with other biomolecules such as DNA and HSA along with DNA cleavage studies. Also, we are planning to go in depth in the release and efficiency as well as in-vivo investigation.
- Since the nanoparticle is prone to be covered with various proteins in circulation, why do the authors only put emphasis on BSA binding study without detecting any other possible protein?
Author response: The comment is nice and because in this work detailed kinetic studies are presented, we used the simplest biomolecules, but work is now undergoing in more detail with other biomolecules such as DNA and HSA along with DNA cleavage studies. Also, we are planning to go in depth into the release and efficiency as well as in-vivo investigation.
- The positive-charged HCQ@CS NPs may have serious safety issues when applied in vivo, how to resolve this potential risk?
Author response: Based on our preliminary work, other related studies and also the work with normal cells besides the cancer cells, we expect to work safely.

Round 2
Reviewer 1 Report
The authors addressed the concerns to some extent. However, the interpretation/discussion is still more than what can truly be said from the obtained results, which is causing leaps in logic from point to point. Careful revision of the experimental design as well as additional data where appropriate are inevitable to provide sufficient evidence behind the statements that the authors make.
1. Please introduce in detail why you studied BSA binding. Nanoparticle always will form protein corona. Some protein will improve the stealth effect of nanoparticles that is good for drug delivery application. There are several ways to make stealth nanoparticles, such as PEGylation, protein coating, cell membrane coating. Here the strategy is like in situ generating protein corona (BSA) for long-circulating nanoparticles. This paper may be useful (https://doi.org/10.1016/j.addr.2023.114895).
2. The introduction is too lengthy. Please shorten it, such as removing unnecessary introduction for lung cancer.
3. Antibacterial application is not necessary. It can not fit into the story.
4. what do you mean "roughly shaped like roadways"? It is curious to describe nanoparticle like roadways. From TEM image, what is the nanoparticle? Please indicate it.
5. Usually, in vitro cytotoxicity of drug-loaded nanoparticle will be low compared with free drug. The authors should explain the reported drug-loaded nanoparticle is more toxic. Improved the functionality like cellular uptake may be a reason (https://doi.org/10.1021/jacs.0c09029). But the evidence should be provided.
6. It is not better to use electrostatic interaction to describe the encapsulation. Electrostatic interaction usually means the interaction between positively charged groups and negatively charged groups. The authors can just simply use noncovalent interactions like hydrogen bond.
Author Response
Reviewer 1: The authors addressed the concerns to some extent. However, the interpretation/discussion is still more than what can truly be said from the obtained results, which is causing leaps in logic from point to point. Careful revision of the experimental design as well as additional data where appropriate are inevitable to provide sufficient evidence behind the statements that the authors make.
Author response: the authors are very much thankful for the reviewers for thorough reviewing the manuscript.
Comment 1. Please introduce in detail why you studied BSA binding. Nanoparticle always will form protein corona. Some protein will improve the stealth effect of nanoparticles that is good for drug delivery application. There are several ways to make stealth nanoparticles, such as PEGylation, protein coating, cell membrane coating. Here the strategy is like in situ generating protein corona (BSA) for long-circulating nanoparticles. This paper may be useful (https://doi.org/10.1016/j.addr.2023.114895).
Author response: Yes, the text has been modified according to the reviewer suggestions
It is crucial to determine whether or not a novel anticancer metalotherapeutic binds to serum transport proteins because intravenous administration is the preferred method for administering such drugs. In fact, examining a novel candidate drug's interactions with significant (the most prevalent) serum proteins is necessary for its pharmacological characterisation. As a result, it is necessary to calculate the binding stoichiometry, affinity constant, and number of (particular) binding sites. Human serum albumin (HAS), immunoglobulin (G), and serum transferrin (Tf) are the three main serum proteins. The medicinal chemistry literature frequently describes interactions between BSA and prospective metallodrugs. This is due to BSA's stability, neutrality in a wide range of metabolic processes, low cost, and structural closeness to its human analogue, HSA [39,40]
[39] Simovic´, AR.; Masnikosa, R.; Bratsos, I.; Alessio, E. Chemistry and reactivity of ruthenium(II) complexes: DNA/protein binding mode and anticancer activity are related to the complex structure. Coordination Chemistry Reviews 2019, 398, 113011-113037. https://doi.org/10.1016/j.ccr.2019.07.008 and references therein.
[40] Wen, P.; Ke, W.; Dirisala, A.; Toh, K.; Tanaka M.; Li, J. Stealth and pseudo-stealth nanocarriers. Advanced Drug Delivery Reviews Volume 198, July 2023, 114895. https://doi.org/10.1016/j.addr.2023.114895
Comment 2. The introduction is too lengthy. Please shorten it, such as removing unnecessary introduction for lung cancer.
Author response: Introduction has been modified and shortened according to the reviewer suggestions.
Comment 3. Antibacterial application is not necessary. It cannot fit into the story.
Author response: Actually, cancer patients are mostly given antibacterial agents in addition to chemotherapeutic agents to prevent various infections. This is because chemotherapy makes cancer patients immunocompromised and susceptible to infections. This is the main reason for testing the antibacterial activity as well. We have mentioned such reason in the introduction.
Comment 4. what do you mean "roughly shaped like roadways"? It is curious to describe nanoparticle like roadways. From TEM image, what is the nanoparticle? Please indicate it.
Author response: Yes, the text has been changed.
Comment 5. Usually, in vitro cytotoxicity of drug-loaded nanoparticle will be low compared with free drug. The authors should explain the reported drug-loaded nanoparticle is more toxic. Improved the functionality like cellular uptake may be a reason (https://doi.org/10.1021/jacs.0c09029). But the evidence should be provided.
Author response: We agree with the reviewer and a text was added.
This may be due to increased functioning like cellular uptake. Proteins make up the second largest portion of cell membranes, and some of them have the ability to mediate cellular uptake, also known as receptor-mediated uptake. When compared to HCQ (1.67 M-1), BSA protein had a higher affinity for HCQ@CS NPs (22.2 M-1), which may be a perfect chance to improve drug targeting and administration. [86,87]. A second reason may be the kinetic stability of the drug-DNA/protein complex as Denny et. al. [88] reported that the cytotoxicity of intercalating agents correlates better with kinetics stability of the drug-DNA/protein complexes than with binding affinity. HCQ@CS NPs-BSA complex showed a higher stability over long tome compared to that of HCQ-BSA as shown in Figure 14.
[86] Li, J.; Kataoka, K. Chemo-physical Strategies to Advance the in Vivo Functionality of Targeted Nanomedicine: The Next Generation. J. Am. Chem. Soc. 2021, 143, 538−559. https://doi.org/10.1021/jacs.0c09029.
[87] Zhang, R.; Qin, X.; Kong, F.; Chen, P.; Pan, G. Improving cellular uptake of therapeutic entities through interaction with components of cell membrane. Drug Deliv. 2019, 26(1), 328–342, doi: 10.1080/10717544.2019.1582730.
[88] Denny, WA.; Wakelin, LPG. Kinetic and Equilibrium Studies of the Interaction of Amsacrine and Anilino Ring-substituted Analogues with DNA. Cancer Research, 1968, 46, 1717-1721.
Figure 14. Kinetic trace of showing the stability of HCQ@CS NPs-BSA adduct over longer time.
Comment 6. It is not better to use electrostatic interaction to describe the encapsulation. Electrostatic interaction usually means the interaction between positively charged groups and negatively charged groups. The authors can just simply use noncovalent interactions like hydrogen bond.
Author response: Yes, we agree and the text was modified according to the reviewer suggestions

Reviewer 2 Report
1) Authors should unify the writing of all abbreviations in the text, particularly FT-IR.
2) The methodological part does not describe sample preparation of CS and CS NPs and conjugates for DLS analysis. Taking into account the low solubility of chitosan, indicated by the authors themselves in response to comments on the absence of NMR spectra, it is simply necessary to describe how the authors were able to prepare a solution of these samples suitable for such an aggregate-sensitive method as DLS.
3) The authors should specify the molecular weight of chitosan.
4) In the methodological part, the authors should describe the protocol of statistical processing of the obtained data.
5) The authors did not provide data on elemental analysis of all objects of the study. This should be done.
6) It is necessary to supplement this article with Rh and Z-potential data measured under biomimetic conditions, taking into account the positioning of the objects obtained by the authors as potential means for their biomedical applications.
Native speaker checking of the text is required.
Native speaker checking of the text is required.
Author Response
Reviewer 2:
Comment 1. Authors should unify the writing of all abbreviations in the text, particularly FT-IR.
Author response: Yes, was corrected
Comment 2: The methodological part does not describe sample preparation of CS and CS NPs and conjugates for DLS analysis. Taking into account the low solubility of chitosan, indicated by the authors themselves in response to comments on the absence of NMR spectra, it is simply necessary to describe how the authors were able to prepare a solution of these samples suitable for such an aggregate-sensitive method as DLS.
Author response: The DLS data were measured in tris-buffer (tris(hydroxymethyl)aminomethane) and in this case, the solubility is enough to do such analysis.
Comment 3: The authors should specify the molecular weight of chitosan.
Author response: Yes, was written.
Comment 4: In the methodological part, the authors should describe the protocol of statistical processing of the obtained data.
Author response: Actually using every technique in this manuscript, we mentioned in the text how we can get the data and what are the equations used and so on…
Comment 5. The authors did not provide data on elemental analysis of all objects of the study. This should be done.
Author response: We have done the elemental analysis as suggested by the reviewer.
Comment 6. It is necessary to supplement this article with Rh and Z-potential data measured under biomimetic conditions, taking into account the positioning of the objects obtained by the authors as potential means for their biomedical applications.
Author response: The zeta potential also was measured in tris-buffer which is of pH equal to 7.4, is nearly the biomimetic pH.
Native speaker checking of the text is required
Round 3
Reviewer 1 Report
One minor comments. The authors claimed "the preparation of chitosan nanoparticles (CNPs) using hydroxychloroquine (HCQ)". This may be not accurate. It is better to say hydroxychloroquine is encapsulated in chitosan nanoparticles (CNPs).
Author Response
Author response to the reviewer’s comments (Manuscript ID: ijms-2559659) entitled Hydroxychloroquine-Loaded Chitosan Nanoparticles Induce Anticancer Activity in A549 Lung Cancer Cells: Design, BSA Binding, Molecular Docking, Mechanistic, and Biological Evaluation
Dear Ms. Winifred Wang, MDPI Office
Thank you for your letter of 29-Aug-2023 and the authors are very much thankful for you and the reviewers for thoroughly reviewing the manuscript. We have taken into consideration all issues mentioned in the reviewers' comments carefully and have revised the manuscript accordingly (in yellow color).
Specific responses to the inquiries are given below: (in blue color)
Reviewer 1: one minor comment. The authors claimed "the preparation of chitosan nanoparticles (CNPs) using hydroxychloroquine (HCQ)". This may be not accurate. It is better to say hydroxychloroquine is encapsulated in chitosan nanoparticles (CNPs)
Author response: The authors are very much thankful for the reviewers for thorough reviewing the manuscript for the third time and we agree with the reviewer and the text was corrected.
Reviewer 2 Report
The authors provided their responses to comments on the text and layout of the manuscript. The article can be published after the grammar of the text has been checked by a native speaker.
The article can be published after the grammar of the text has been checked by a native speaker.